



Variations of the 630.0 nm airglow emission with meridional
neutral wind and neutral temperature around midnight
Chih-Yu Chiang[1], Sunny Wing-Yee Tam[1], Tzu-Fang Chang[1,2]
[1] Institute of Space and Plasma Sciences, National Cheng Kung University, Tainan
70101, Taiwan
[2] Institute for Space-Earth Environmental Research, Nagoya University, Nagoya
464-8601, Japan



**Abstract**

9         Enhancements in 630.0 nm airglow around midnight at equatorial latitudes were

observed by many optical observations. Such features had been suggested as the
signature of thermospheric midnight temperature maximum (MTM) effect, which was
associated with temperature and meridional neutral winds. This study investigates the
influence of neutral temperature and meridional neutral wind on the volume emission
rates of the 630.0 nm nightglow. We utilize the SAMI2 model to simulate the charged
and neutral species at the 630.0 nm nightglow emission layer under different
temperatures with and without the effect of neutral wind. The results show that the
neutral wind is more efficient than temperature variation in affecting the nightglow
emission rates. However, the emission rate features a local maximum in its variation
with the temperature. Two kinds of tendencies can be seen regarding the temperature
that corresponds to the turning point, which is named the turning temperature ($T_t$) in
this study: firstly, $T_t$ decreases with the emission rate for the same altitude; secondly,
for approximately the same emission rate, $T_t$ increases with the altitude.





**1. Introduction**

The atomic oxygen red line at 630.0 nm is the most prominent emission in the

nighttime ionosphere. It usually forms an emission layer in the F region at altitudes of
~200–300 km and can be easily observed from ground-based observatories or
satellites [*Nelson and Cogger*, 1971; *Kelley et al.*, 2002; *Thuillier et al.*, 2002]. The
emission is related to $O(^1D)$, whose production in the nighttime is mainly via the
charge exchange and dissociative chemical processes listed as follows:

$$O^+ + O_2 \rightarrow O_2^+ + O \qquad\qquad (1)$$

$$O_2^+ + e^- \rightarrow O(^1D) + O \qquad\qquad (2)$$

$$O(^1D) \rightarrow O(^3P) + h\nu(630.0\ nm) \qquad\qquad (3)$$

Based on the $[O^+] \sim N_e$ (electron density) approximation [*Peterson et al.*, 1966;

*Link and Cogger*, 1988] in the F2 region, the intensity of the $OI(^1D)$ 630.0 nm spectral
line is usually used to identify the ionospheric electron density variations. From a rich
history in the literature,  the intensity of $OI(^1D)$ 630.0 nm airglow emissions is
known as Midnight Brightness Wave (MBW) [*Herrero and Meriwether*, 1980;
*Herrero et al.*, 1993; *Colerico et al.*, 1996; *Colerico and Mendillo*, 2002].

During occurrences of MBW, enhancements in temperature are usually

observed around local midnight, which are termed Midnight Temperature Maximum
(MTM) effect. *Harper* [1973] and *Spencer et al.* [1979] first reported the MTM



phenomenon. The cases in their studies were observed by the incoherent scatter radar
from Arecibo and the NATE experiment aboard the Atmospheric Explorer E (AE-E)
satellite, respectively. The amplitude of the temperature bulge was found to range
from 20 to 200 K [*Spencer et al.*, 1979; *Burnside et al.*, 1981; *Colerico and Mendillo*,
2002; *Meriwether et al.*, 2008]. In addition, a number of studies about midnight
brightness have reported the relation between *in-situ* temperature and neutral wind
measurements [e.g., *Herrero and Meriwether*, 1980; *Sastri et al.*, 1994, *Colerico et al.*,
1996, 2002; *Otsuka et al.*, 2003; *Mukherjee et al.*, 2006]. *Adachi et al.* [2010] showed
a 14-day time span of airglow observations obtained from the Asian sector by the
Imager of Sprites and Upper Atmospheric Lightning (ISUAL) [*Chang et al.*, 2012;
*Chiang et al.*, 2013] on board the FORMOSAT-2 satellite. On the basis of the
observation time and location, they suggested that the equatorial airglow probably
corresponded to the midnight brightening wave (MBW) which is in association with
the occurrence of MTM. Furthermore, *Chiang et al.* [2013] statistically investigated
the global midnight brightness according to seasons and found that the global
midnight brightness near the equatorial regions was controlled by different
mechanisms. In the study, the features and behavior of the 630.0 nm midnight
intensity were investigated by analyzing the optical images obtained by ISUAL
(Supplement I). Cases of global midnight brightness were successfully categorized





into four types that were mainly due to the influence of temperature changes, neutral
wind and ionospheric anomaly.

Based on the previous studies, it is known that temperature and meridional

neutral wind are correlated and associated with manifestations of MTM. Thus, we
want to discuss these two effects at the same time. In this study, we calculate the volume
emission rates to understand the influence of neutral temperature and meridional
neutral wind on the 630.0 nm nightglow. We shall discuss the sensitivities of the
emission rates to the temperature and the densities of several neutral and charged
species. Moreover, some new features will also be shown in the discussion section.

**2. Model features**

Temperature changes and meridional neutral wind can influence the O($^1$D)

nightglow intensity through particle densities. The volume emission rate of the 630.0
nm nightglow in the F2 region [*Sobral et al.*, 1993] can be   derived from the
chemical process of 630.0 nm nightglow (Supplement II). It is shown as follows:
$$I_{630} = \frac{A_{1D}\mu_D\gamma[O_2][O^+]}{k_1[N_2] + k_2[O_2] + k_3[O] + A_{1D} + A_{2D}} \; , \qquad (4)$$
where $\mu_D$ is the quantum yield of O($^1$D), which is about 1~1.3 [*Torr and Torr*, 1982]; $\gamma$
is the rate coefficient of Reaction (1) [*St.-Maurice and Torr*, 1978]; $k_1$, $k_2$ and $k_3$ are
the rate coefficients of O($^1$D) quenched by N$_2$, O$_2$ and O, respectively [*Langford et al.*,



1986; *Streit et al.*, 1976; *Sun and Dalgarno*, 1992]; and $A_{1D}$ and $A_{2D}$ are the transition
coefficients [*Froese-Fischer and Saha*, 1983]. The formulas for the rate coefficients
[*Vlasov et al.*, 2005] are listed in Table 1. The production rate of $O(^1D)$ is contributed
by the oxygen ion density $[O^+]$ and the molecular oxygen density $[O_2]$ through the
linked reactions (1) and (2). The major loss rates of $O(^1D)$ are associated with the
densities of molecular oxygen $[O_2]$, molecular nitrogen $[N_2]$, and atomic oxygen $[O]$,
as reflected in Eq. (4). The densities $[O^+]$, $[O_2]$, $[N_2]$ and $[O]$ and the rate coefficients
$\gamma$, $k_1$, $k_2$ and $k_3$ all depend on temperature. In addition, $[O^+]$ may change with the
neutral wind conditions. In order to determine $I_{630}$ under different temperatures and
neutral wind conditions, one must first determine the densities of the relevant species.
In this study, $[O^+]$ and plasma temperatures under various conditions are found by the
SAMI2 model of the Naval Research Lab [*Huba et al.*, 2000]. SAMI2 is a two-
dimensional, first-principle model of the comprehensive low to mid-latitude
ionosphere. SAMI2 code includes most of the mechanisms that should be considered
in the ionosphere. There are photoionizations, chemical process, effects by the
magnetic and electric fields, plasma dynamics and the influence from the neutral
atmosphere. The input variables, neutral species, are specified using the empirical
codes, the Mass Spectrometer Incoherent Scatter model (NRLMSISE-00) [*Picone et*
*al.*, 2002] for neutral densities and the Horizontal Wind Model (HWM-93) [*Hedin et*



*al.*, 1996] for neutral wind. The continuity and momentum equations of seven ion
species ($H^+$, $He^+$, $N^+$, $O^+$, $N_2^+$, $NO^+$, and $O_2^+$) are solved in the code.

In order to understand the differences due to the meridional neutral wind, we

apply the SAMI2 model with and without neutral wind by changing the multiplicative
factor of neutral wind (tvn0) to see the differences between two solstices. Thus, we
simulate the cases of February 1, 2007 (northern winter) and August 1, 2007 (northern
summer). In the simulations, we suppose that the solar and geomagnetic activities are
in quiet conditions (F10.7 index = 60, Ap index = 7). The simulations are run for the
altitude range between 150 and 1000 km from -30° to +30° geomagnetic latitudes.
Inside this region, we use 100 geomagnetic field lines and 201 grid points along the
field line. Our report of the results will focus on the locations at -5° and +5°
geomagnetic latitude (+2° and +12° geographic latitude respectively) along the 100°E
geographic longitude, which intersects these latitudes in the Asian region. Figure 1
shows the $O^+$ density along the magnetic lines with apex altitudes between 265 and
315 km in the latitude-altitude plane at the time and longitude described above. Figure
1(a) shows the results under the condition that lacks neutral wind, and Fig. 1(b) shows
the results with the effect of normal neutral wind. The two left panels are for February
1, 2007 and the two right panels are for August 1, 2007. The arrows plotted in Fig. 1(b)
indicate the strength and directions of the meridional neutral wind. Comparison of Fig.



1(a) and 1(b) clearly shows that meridional winds transport the plasma along the
magnetic field line and change the plasma density distribution. And this change of the
plasma profile could directly modify the emission rate in Eq. (4). The dashed lines,
which correspond to $\pm 5°$ geomagnetic latitude, indicate the locations where the
intensity of the 630.0 nm nightglow is examined in detail in this study.

**3. Results and Analysis**
Based on Eq. (4), $I_{630}$ under different temperatures and different neutral wind
conditions is plotted in Fig. 2. The neutral wind conditions for the results in Fig. 2 are
the same as those   for Fig. 1. The strength and directions of the neutral winds are
indicated by the arrows shown in Fig. 1. The simulation results shown in the figure
are for (a) February 1, 2007 and (b) August 1, 2007, with the left and right panels
respectively corresponding to -5° and +5° geomagnetic latitude. The letters, A, B, C,
D and E, indicate the altitudes of 220, 230, 240, 250 and 260 km, respectively. The
dotted lines indicate the results with normal neutral wind effect; the solid lines
indicate the results without neutral wind effect. Note that the temperatures of around
650°K, corresponding to the leftmost points of the lines in the figure, were the initial
neutral temperatures obtained from the NRLMSISE-00 model at the various altitudes.
These neutral temperatures are input into the SAMI2 model, and we set up the



48-hour data as a running loop to obtain the plasma data. For different temperature
conditions, we increase 50 K in the input temperature every time via modifying the
codes of SAMI2 and ran the simulations to calculate new values for the emission
intensity.
From Fig. 2, we can see the influence of temperature and neutral wind on the
nightglow emission. Note that the neutral wind conditions are as in Fig. 1: Fig. 1(a)
for without wind condition and the Fig. 1(b) for normal wind condition. The influence
of the temperature variations on $I_{630}$ is usually less than 3 photons/cm$^3$/sec at the
heights of 220 to 260 km. The variation of $I_{630}$ with temperature, however, is not
monotonic; there is a maximum in the intensity as the temperature changes. In terms
of height, as $I_{630}$ depends on the local neutral and charged particle densities in
accordance with Eq. (4), the emission is the strongest at 230 km, except for the
condition of very weak emission (< 1 photon/cm$^3$/sec) that occurs at +5° geomagnetic
latitude in August with normal wind effect (right panel of Fig. 2(b)).
As for the influence of the neutral wind on February 1, 2007 of Fig. 2(a), both
locations (±5° geomagnetic latitude) clearly feature significantly smaller $I_{630}$ under
this effect. We suggest that this is due to the meridional neutral wind blowing
equatorward in both hemispheres (see Fig. 1) and pushing the plasma upward along
the field lines, reducing the local charged particle densities and consequently the



emission rates as well. On August 1, 2007 of Fig. 2(b), the neutral wind causes the
intensity at +5° geomagnetic latitude to decrease significantly for the same reason as
the wind direction is locally southward (equatorward). This equatorward neutral wind,
however, has an opposite effect on the intensity at -5° geomagnetic latitude; being
locally poleward, the wind pushes the plasma downward along the field lines,
increasing the local charged particle densities and consequently the emission rates as
well.
From Eq. (4), we can see that $I_{630}$ is related to the densities of several neutral
species as well. In order to find out how the temperature affects the overall chemical
process that leads to the 630.0 nm emission, a few profiles of relevant parameters as
functions of temperature in Fig. 3, based on the condition at 230 km altitude and -5°
geomagnetic latitude on February 1, 2007. In Fig. 3(a), the O($^1$D) loss-rate terms
associated with [O], [N$_2$] and [O$_2$], which are shown in dotted, dashed and solid lines
respectively. The value of  $\gamma$ [O$^+$][O$_2$], which is related to the O($^1$D) production rate
and is in the numerator of Eq. (4), corresponding to Fig. 3(b). The dotted line
represents the normal neutral wind condition, and the solid line for the windless
condition.

**4. Discussion**



From Fig. 1(a), we can see that along the field lines, the $O^+$ density is maximum
around the geomagnetic equator when there is no neutral wind, whether it is in the
summer or winter season. But the $[O^+]$ maxima tilt to the winter hemisphere in the
presence of summer-to-winter neutral wind at the geomagnetic equator, as shown in
Fig. 1(b). The density profiles of the charged particles along the field lines are clearly
influenced by the neutral wind.
From the results that include the normal wind effect as shown in Fig. 2, the
intensities on opposite sides of the geomagnetic equator are very different. The
weaker emission is in the summer hemisphere, and brightness of higher intensity
appears in the winter hemisphere. In previous studies, *Rishbeth and Setty* [1961]
found that NmF2 was larger in winter than in summer, and they first suggested the
possibility of composition change being the cause of the winter anomaly. *Rishbeth*
[1972] and *Torr and Torr* [1973] suggested that the anomaly might be due to
transequatorial neutral wind blowing from the summer hemisphere to the winter
hemisphere. Therefore, the enhancement of the emission at the low latitudes of the
winter hemisphere should be the results of plasma accumulation caused by the neutral
wind effect.
Figure 2 shows the influence of temperature and neutral wind on the nightglow
emission rates. We estimate the intensity change under different neutral wind



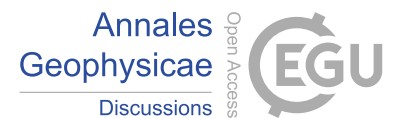

conditions based on the location at 230 km altitude and -5° geomagnetic latitude on
February 1, 2007. In this situation, the emission would be reduced by the wind flow,
and the average change is about 0.690 photon/cm$^3$/sec for every m/sec of the wind
speed. In comparison, the change due to temperature variation is just 0.015
photon/cm$^3$/sec for every K. The ratio of the two numbers is 46. Consideration of
other conditions may reduce the corresponding ratio, but it should still be at least 20.
According to earlier studies, the neutral wind speed is generally 0-300 m/sec in the F
region [*Dyson et al.*, 1997], while the amplitude of the temperature bulge due to the
MTM effect has been found to range from 20 to 200 K [*Burnside et al.*, 1981;
*Colerico and Mendillo*, 2002]. Even if one assumes the maximum wind speed is just
60 m/sec as in the simulations in this study, it would require a temperature change of
1200 K to match the same change in emission intensity caused by the neutral wind.
Such a large temperature change is not realistic in comparison with the maximum
observed difference of 200 K. Thus, the emission rate of nightglow, realistically, is
influenced more by the neutral wind than temperature change when the former
mechanism is clearly present.

Previously Chiang et al. [2013] examined the occurrence rates of global midnight

brightness observed by FORMOSAT-2/ISUAL. In order to verify the enhancement of
the emission intensity in the winter hemisphere by the neutral wind, we examined the

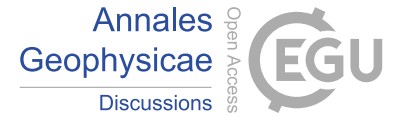

FORMOSAT-2/ISUAL data that correspond to the specific regions and seasons
considered in our simulations and the results are shown in Fig. 4(a) and (b). We found
that among the 22 valid observation days during January and February, ~77% of the
days featured the appearance of nightglow bright spots in the low-latitude region of
the winter hemisphere (Fig. 4(a)). Furthermore, ~83% of the 30 valid observation
days during July-August also featured nightglow bright spots at low latitudes in the
corresponding winter hemisphere (Fig.4(b)). Thus, statistical results regarding the
location of nightglow bright spots agree with the simulation results that demonstrate
the crucial role of the neutral wind in affecting the location of high-intensity
nightglow regions.

The densities and some of the rate coefficients are temperature dependent, as

given in Eq. (4). We analyze the change with temperature of the individual terms in
Eq. (4) change with temperature. In Fig. 3(a) and Fig. 3(b), we shown the terms in the
numerator and denominator on the right-hand side of Eq. (4) and found that all these
terms increase with temperature. However, if we consider the derivative of the terms
with respect to temperature, which characterizes how sensitive the terms are to
temperature change, we notice that the derivatives for $k_1[N_2]$ and $k_3[O]$ increase with
temperature while those for $k_2[O_2]$ and $\gamma [O^+][O_2]$ decrease, as shown in Fig. 3(a)
and 3(b). How the variations of these terms affect the dependence of $I_{630}$ on



temperature can now be understood from the right-hand side of Eq. (4). In particular,
the numerator, which characterizes the production rate of $O(^1D)$ and is proportional to
$\gamma\,[O^+][O_2]$, increases with temperature while featuring a relatively large increase at
lower temperatures (less than ~750 K). On the other hand, the denominator, which
characterizes the total loss rate of $O(^1D)$ and is dominated by $k_1[N_2]$ as Fig. 3(a)
indicates, features a relatively large increase at higher temperatures (larger than ~750
K). Upon division of the numerator by the denominator, the plot of $I_{630}$ vs.
temperature is thus characterized by quasi-parabolic lines with the presence of a local
maximum --- or a turning point in the curve --- as shown in Fig. 2. We refer to the
temperature that corresponds to such a local maximum as the turning temperature ($T_t$).
Below $T_t$, $I_{630}$ increases with temperature, meaning that the increase in the production
of $O(^1D)$ associated with a rise in the temperature is more efficient than the increase in
its loss. In contrast, $I_{630}$ decreases with temperature above $T_t$, meaning that the
increase in the production of $O(^1D)$ associated with a rise in the temperature is less
efficient than the increase in its loss. Thus, $T_t$ has the significance of being the
temperature at which the production and loss rates of $O(^1D)$ are equally sensitive to a
temperature change.

Figure 5 shows a plot of $T_t$ versus the emission rate $I_{630}$ at specific altitudes. The

results include all the cases shown in Fig. 2 with different symbols indicating different





altitudes. Two kinds of tendencies can be seen from the plot: firstly, $T_t$ decreases with
$I_{630}$ for the same altitude; secondly, for approximately the same emission rate, $T_t$
increases with the altitude. This is the first result to show these tendencies of the
turning temperature.

Observations of the movement of MTM temperature bulge and that of nightglow

have led to postulations of an association between pressure bulge and nightglow
intensity [*Colerico et al.*, 1996; *Colerico and Mendillo*, 2002; *Meriwether et al.*,
2008]. However, the high intensities of the observed nightglow have not been
successfully reproduced using existing models incorporating the MTM effect, such as
the NCAR thermosphere-ionosphere-electrodynamic general circulation model
(TIEGCM), as pointed out by *Colerico and Mendillo* [2002] and *Meriwether et al.*
[2008]. Note that temperature was not included as a varying quantity in traditional
ionospheric models. Thus the simulation study of temperature effect upon nightglow
intensity is lacking. Our simulation results have demonstrated the unexpectedly
non-monotonic dependence of the intensity of nightglow on the neutral temperature,
with the turning temperature $T_t$ that arises from the dependence implying a limitation
for the growth of the emission rates. As the temperature increases above $T_t$, the
emission rates do not continue to grow. In fact, temperature change such as in the case
of heat transfer is affected by the density, which controls the heat capacity. At the





same time, temperature change may generate pressure difference and lead to transport
that changes density profiles. As nightglow intensity depends also on particle densities,
its non-monotonic variations with temperature are in fact due to the combination of
temperature and density. While our study suggests that neutral wind is the dominant
drive of the $I_{630}$ variation, its influence, however, is via transportation of plasma and
neutral particles, in which case consideration of the effect of temperature on the
density is essential. Moreover, it has not been established that MTM is affected by the
wind primarily. The combination of temperature and density, which has shown to
cause non-monotonic results in this study, may very well be an important factor in the
study of MTM. Thus, if one wants to fully reproduce the observation results, we
suggest other extra factors associated with temperature variations should also be
considered, such as different tidal modes from lower atmosphere [*Akmaev et al.*,
2009]. Our findings of the turning temperature tendencies can help as a guide for
choosing the background temperature in future modeling attempts to obtain intensities
of nightglow brightness comparable to those observed from ground or from space.

Shepherd [2016] investigates the possible extent of the MTM at ~ 20°N–40°N,

considering O(1D) airglow volume emission rates, Doppler temperatures, and neutral
(zonal and meridional) observations by the Wind Imaging Interferometer (WINDII)
experiment on board the Upper Atmosphere Research Satellite (UARS). Their results



provide us the relations of the zonal wind to the O(1D) emission rate and of the
meridional wind to the temperature. Thus, it potentially leads us to a more extensive
future study in simulation to reproduce the observation and statistical results provided
by Shepherd in 2016.

**5. Conclusion**

Previous studies of the MTM effect have pointed out that the temperature

anomaly influences the nighttime behavior of the thermosphere. And the neutral wind
also plays a key role to cause the intensity variations in the nighttime ionosphere.
Based on our simulation results, both temperature change and meridional neutral wind
could cause the 630.0 nm nightglow intensity to vary while the latter is more effective.
An unexpected aspect of the results is the non-monotonic dependence of the emission
rate on temperature, featuring a turning point as the temperature changes. The
temperature $T_t$ at which the turning point occurs corresponds to a balanced condition
between the production and loss of $O(^1D)$. Thus, our results help understand how the
overall chemical process of nightglow is affected by the variations of neutral
temperature and neutral wind. Two kinds of tendencies can be seen regarding the
turning temperature $T_t$. One is the higher $T_t$ corresponding to higher altitude at the
same emission rate, the other is the higher $T_t$ corresponding to lower emission rate at



the same altitude. Our findings of these turning temperature tendencies can guide
future modeling attempts to match the observed nightglow brightness intensities.

**Acknowledgements**

The authors acknowledge the FORMOSAT-2/ISUAL science and operator team

to provide image data (http://sprite.phys.ncku.edu.tw/en/about-cdf-distribution). The
work by C. Y. Chiang and S. W. Y. Tam is supported by Taiwan's Ministry of Science
and Technology grants MOST105-2111-M-006-007. T. F. Chang acknowledges
support by the Ministry of Education, Taiwan R.O.C., from The Aim for the Top
University Project to National Cheng Kung University.













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





Table 1. Reactions and rate coefficients related to the volume emission rate of the
630.0 nm airglow

| Reactions | Rate Coefficients ($cm^3s^{-1}$, $s^{-1}$) |
|---|---|
| $O^+ + O_2 \rightarrow O_2^+ + O$ | $\gamma = 2.82\times10^{-11} - 7.74\times10^{-12}(T_{eff}/300) + 1.07\times10^{-12}(T_{eff}/300)^2 - 5.17\times10^{-14}(T_{eff}/300)^3 + 9.65\times10^{-16}(T_{eff}/300)^4$ |
| $O(^1D) + N_2 \rightarrow O + N_2$ | $k_1 = 2\times10^{-11}\exp(107.8/T_n)$ |
| $O(^1D) + O_2 \rightarrow O + O_2$ | $k_2 = 2.9\times10^{-11}\exp(67.5/T_n)$ |
| $O(^1D) + O \rightarrow O + O$ | $k_3 = (3.73 + 1.1965\times10^{-1} T_n^{0.5} - 6.5898\times10^{-4} T_n)\times10^{-12}$ |
| $O(^1D) \rightarrow O + h\nu(630.0nm)$ | $A_{1D} = 7.1\times10^{-3}$ |
| $O(^1D) \rightarrow O + h\nu(634.4nm)$ | $A_{2D} = 2.2\times10^{-3}$ |

Note: $T_{eff} = 0.67T_i + 0.33T_n$ ($T_{eff}$: effective temperature, $T_i$: ion temperature, $T_n$: neutral temperature)
[*St.-Maurice and Torr*, 1978]















**Figure Captions**
Figure 1. Oxygen ion density plotted in the latitude-altitude plane at 23:00 LT on

February 1, 2007 (left panels) and August 1, 2007 (right panels) in the Asian

region (100°E longitude) from the SAMI-2 model: (a) without neutral wind; (b)

with the effect of normal neutral wind, whose strength and directions are

indicated by the arrows.

Figure 2. The results of 630.0 nm emission rate at 23 LT at different temperatures and

under different neutral wind conditions for (a) February 1, 2007 and (b) August

1, 2007: left and right panels respectively for -5° and +5° geomagnetic latitude;

the letters, A, B, C, D and E, for the altitudes of 220 km, 230 km, 240 km, 250

472          km and 260 km, respectively; for normal neutral wind effect (black dotted lines)

and windless conditions (red solid lines). The neutral wind conditions of Fig. 2

are the same as those shown in Fig. 1.

Figure 3. Profiles of the terms in Eq. (4) that are associated with neutral and charged

species versus temperature, based on 230 km altitude and -5° geomagnetic

latitude on February 1, 2007, with and without neutral wind: (a) the loss-rate

terms associated with [O], [N$_2$] and [O$_2$]; (b) the production-rate term $\gamma$

[O$^+$][O$_2$].

Figure 4.   FORMOSAT-2/ISUAL data in the specific regions and seasons considered





in our simulations: (a) Among the 22 valid observation days during

January-February, ~77% of the days featured the appearance of nightglow

bright spots in the low-latitude region of the winter hemisphere; (b) About 83%

of the 30 valid observation days during July-August also featured nightglow

bright spots at low latitudes in the corresponding winter hemisphere.

Figure 5. Plots of the emission rates against the turning temperature between 220-260

487        km altitudes.

















Figure 1

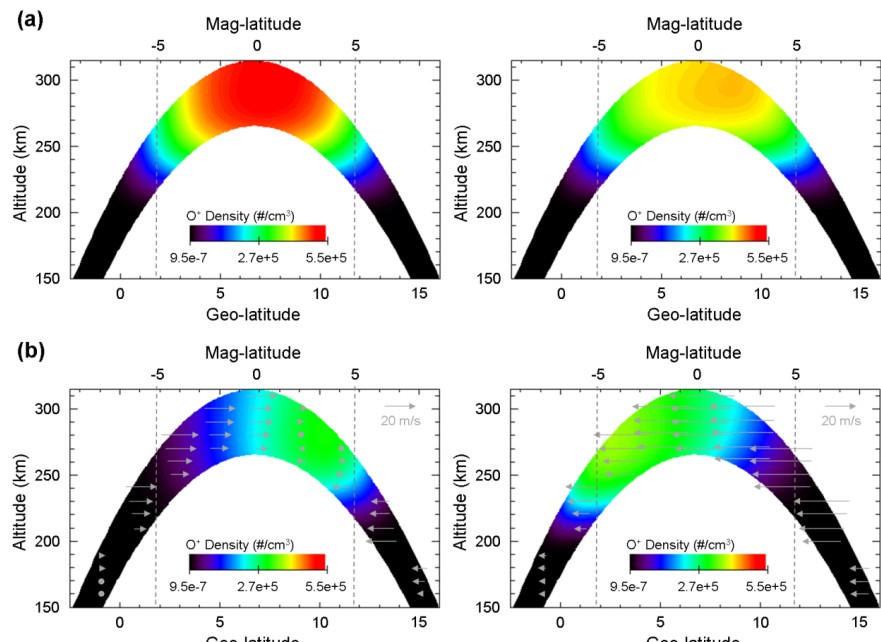















Figure 2

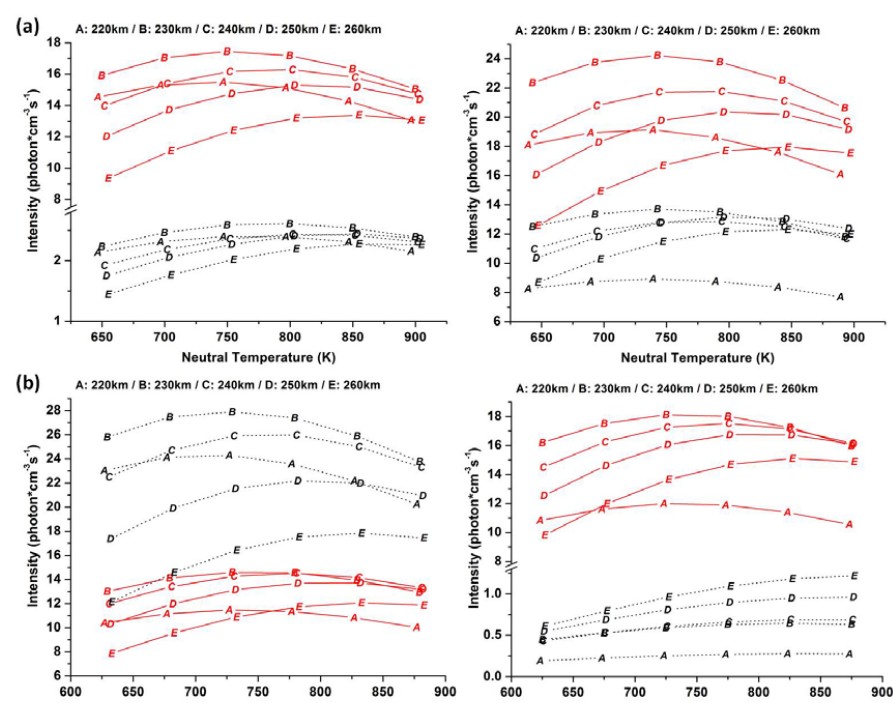















Figure 3

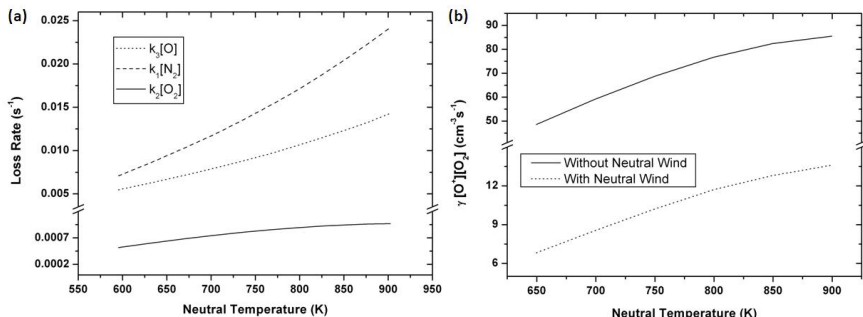
















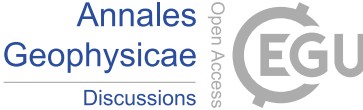



Figure 4

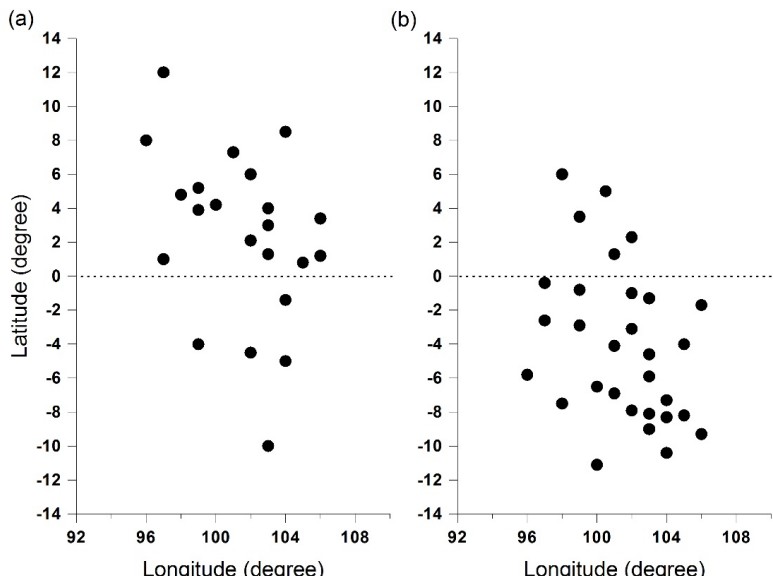
















Figure 5

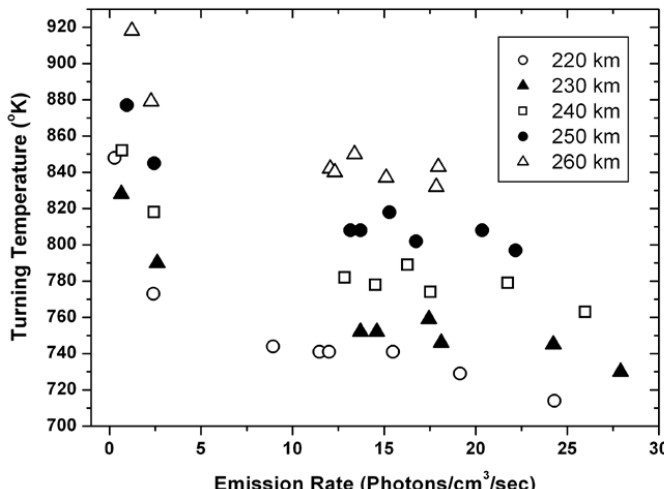
