# Peer review of "Variations of the 630.0 nm airglow emission with meridional neutral wind and neutral temperature around midnight"

_Annales Geophysicae, 2018_

## Referee Comment (RC1) · Anonymous Referee #1 · 27 Feb 2018

In this paper, volume emission rate of the 630-nm airglow is calculated using the SAMI2 model, which is a numerical model of the ionosphere. The authors investigate effects of the neutral winds and temperatures on the volume emission rate, but their argument is still only qualitative. This reviewer considers that quantitative investigation is needed. Therefore, major revision is needed before its publication.

Although the authors describe that effect of the meridional neutral wind is dominant, it is obvious from the equation of the volume emission rate because the volume emission rate is proportional to a product of the plasma and atomic oxygen densities. Meridional neutral winds move the plasma along the magnetic field line and modify plasma density

distribution. Consequently, effects of the neutral winds is dominant.

This reviewer recommends the author to calculate the 630-nm airglow intensity by integrating the volume emission rate along the altitude, and show it as a function of the neutral temperature and meridional neutral winds. The, the authors should argue quantitatively how much the neutral temperature affect the 630-nm airglow intensity compared to the effects of the neutral winds.

Minor comments:

- Figure 1: Arrows representing wind velocity is not seen clearly. - L. 916, Figure 2 –> Figure 3

---

## Referee Comment (RC2) · Anonymous Referee #2 · 7 Mar 2018

Chiang et al. demonstrate the influence of meridional wind and neutral temperature to the intensity of 630.0 nm nightglow around the equatorial midnight, altering the SAMI2 model for the resulting plasma density and temperature with the inputs from NRLMSISE-00 for neutral densities and the HWM-93 for neutral wind vectors. The work is potentially interesting and novelty to the community, particularly the finding with respect to the neutral temperature. However, the literature survey by the authors seems to be hasty, the major lacking is that the role of meridional wind to the midnight 630.0 nm airglow enhancement seeing by ISUAL Imager has been studied and published (Rajesh et al.(2014) doi 10.1002/2014JA019927). In addition, the manuscript requires an editing for English before it can be published in the peer-review journal. Given the

interesting result and a very valuable dataset, I encourage the authors in extending the content in greater detail that be able to deliver the science finding clearly. Please see further comment below. Summary: Consider for publication after substantial revision Major points: (1) Observation data Since the satellite data are used, it would be appropriate to cite Frey et al.(2016) (doi 10.1002/2016JA022616) for the instrument details and the first results of the limb imaging of 630.0 nm airglow using ISUAL by Rajesh et al. (doi 10.1029/2009JA014087 ). The authors put the observation data in the Supplement for some reasons, but it could be nicer if move the section to the main content. The observation data deserve more attention and discussion. (2) The effect of meridional winds to the 630.0 nm midnight brightness By reading this work and Rajesh et al. (2014), I happened to find many similarities in between. Both of the groups modulate the HWM-93 meridional winds on the SAMI2 model and apparently find that the meridional wind utilizes the location and intensity of the airglow brightness. What is the novelty of this work out of Rajesh et al. (2014) in the effect of meridional winds to the midnight brightness? The authors should include the comparison in the content and give the credit to the previous work properly. Line 116-117 What is special of O+ density along the magnetic line with apex altitude between 265 and 315 km ? Can you show the model result between altitude 150 to 315 km for all latitude? Line 214-226 Again, what is the new finding out of fig.3 in Rajesh et al. (2014) ? Figure 1 has to be modified, what is the reason that the authors didn't convert [O+] density to volume emission rate of 630.0 nm nightglow while the observation images are the airglow intensities?

Minor points line by line: Line 43 enhancement > increase Line 45-46 ".…first reported the MTM… " should be "…reported the MTM phenomenon first" Line 61 What are the different mechanisms addressed in Chiang et al. (2013)? The readers would be pleased to learn the relevant work leading by the same author. Line 142-143 Rewrite the sentence please. Line 193-195 Rewrite the sentence please. Line 202-203 Rewrite the sentence please.

End review

6th March, 2018

---

## Author Comment (AC1) · 18 Apr 2018

Comment: In this paper, volume emission rate of the 630-nm airglow is calculated using the SAMI2 model, which is a numerical model of the ionosphere. The authors investigate effects of the neutral winds and temperatures on the volume emission rate, but their argument is still only qualitative. This reviewer considers that quantitative investigation is needed. Therefore, major revision is needed before its publication.

Answer: We would like to thank Referee #1 for reading our article carefully and providing us helpful and valuable suggestions for improving our manuscript. About quantitative investigation, we have added new figures and addressed it in detail in our

manuscript. We have also revised the manuscript accordingly by taking into account the Referee's comments. We hope that Referee #1 now finds the manuscript acceptable for publication.

Comment: Although the authors describe that effect of the meridional neutral wind is dominant, it is obvious from the equation of the volume emission rate because the volume emission rate is proportional to a product of the plasma and atomic oxygen densities. Meridional neutral winds move the plasma along the magnetic field line and modify plasma density distribution. Consequently, effects of the neutral winds is dominant. This reviewer recommends the author to calculate the 630-nm airglow intensity by integrating the volume emission rate along the altitude, and show it as a function of the neutral temperature and meridional neutral winds. The, the authors should argue quantitatively how much the neutral temperature affect the 630-nm airglow intensity compared to the effects of the neutral winds.

Answer: Thanks for Referee #1's nice suggestion, we have added the suggested quantitative investigation in our manuscript as follows: In order to quantitatively describe the effects of neutral temperature and meridional neutral winds, we calculate the 630-nm airglow intensity by integrating the volume emission rate along the altitude. So we make two new plots [Fig. (a) and Fig. (b)] to show how the integrated emission rates vary with the increasing neutral temperature and neutral winds, respectively.

Fig. (a) shows the result regarding the integrated emission rate as affected by neutral temperature (at -5° geomagnetic latitude on February 1, 2007). The curve in red is fitted as 2nd-order polynomial : $\Delta I = (0.1354 \pm 0.0069) \times \Delta T - (4.6835E(-4) \pm 2.6524E(-5)) \times (\Delta T)^2$ where $\Delta I$ (/cm^3*s) is the change in integrated emission rate and $\Delta T$ (K) is the increase in neutral temperature, compared with the standard conditions of 650 K neutral temperature and zero neutral wind.

Fig. (b) shows the result regarding the integrated emission rate as affected by neutral wind. The results are obtained based on the same standard conditions

as those considered in Fig. (a). The curve in red fits an exponential function : $\Delta I=(64.8883\pm0.7772)\times(1-\exp(-(0.0885\pm0.0041)\times\Delta W))$ , where $\Delta I$ (/cm^3*s) is the change in integrated emission rate and $\Delta W$ (m/s) is the change in neutral wind velocity.

Therefore, we combine the results of the two fitting functions to approximate the overall change in the integrated emission rate due to the two effects: $\Delta I=0.1354\times\Delta T-4.6835E(-4)\times(\Delta T)^2+64.8883\times(1-\exp(-0.0885\times\Delta W))$ Based on the function, we can quantitatively compare the neutral temperature effect with the neutral wind effect. In Fig. (a), the maximum change of the integrated emission rate by increasing the neutral temperature is 9.7859 (/cm3*s) at 145 K. To get the same changes of the emission rate by varying the neutral wind, it just requires a neutral wind velocity of 1.85 m/s. Above such a velocity, the neutral wind effect would certainly be larger than that of the neutral temperature for this case.

Comment: Minor comments: - Figure 1:Arrows representing wind velocity is not seen clearly.

Answer: We have replotted the figure as follows (Fig. (c)), thank you.

Comment: - L. 916, Figure 2 –> Figure 3

Answer: Our manuscript does not have Line 916. We searched all the "Figure 2" in our article but did not find a similar typo as mentioned. If Referee #1 can still identify the typo, please let us know again. We would revise it. Thank you.

Please also note the supplement to this comment:
https://www.ann-geophys-discuss.net/angeo-2018-5/angeo-2018-5-AC1-supplement.pdf

———————————————————

[Figure]

[Figure]

**Fig. 1.** Figure (a)

[Figure]

$$\Delta I = 64.8883 \times (1 - \exp(-0.0885 \times \Delta W))$$

**Fig. 2.** Figure (b)

[Figure]

**(a)**

Mag-latitude

Altitude (km)

O⁺ Density (#/cm³)

9.5e-7    2.7e+5    5.5e+5

Geo-latitude

**(b)**

Mag-latitude

Altitude (km)

20 m/s

O⁺ Density (#/cm³)

9.5e-7    2.7e+5    5.5e+5

Geo-latitude

**Fig. 3.** Figure (c)

---

## Author Response (AR1)

In this paper, volume emission rate of the 630-nm airglow is calculated using the SAMI2 model, which is a numerical model of the ionosphere. The authors investigate effects of the neutral winds and temperatures on the volume emission rate, but their argument is still only qualitative. This reviewer considers that quantitative investigation is needed. Therefore, major revision is needed before its publication.

We would like to thank Referee #1 for reading our article carefully and providing us helpful and valuable suggestions for improving our manuscript. About quantitative investigation, we have added new figures (Fig. 4) and addressed it in detail in our manuscript. We have also revised the manuscript accordingly by taking into account the Referee's comments. We hope that Referee #1 now finds the manuscript acceptable for publication.

Although the authors describe that effect of the meridional neutral wind is dominant, it is obvious from the equation of the volume emission rate because the volume emission rate is proportional to a product of the plasma and atomic oxygen densities. Meridional neutral winds move the plasma along the magnetic field line and modify plasma density distribution. Consequently, effects of the neutral winds is dominant. This reviewer recommends the author to calculate the 630-nm airglow intensity by integrating the volume emission rate along the altitude, and show it as a function of the neutral temperature and meridional neutral winds. The, the authors should argue quantitatively how much the neutral temperature affect the 630-nm airglow intensity compared to the effects of the neutral winds.

Thanks for Referee #1's nice suggestion, we have added the suggested quantitative investigation in Line 243-268 as follows:

In order to quantitatively describe the effects of neutral temperature and meridional neutral winds, we calculate the 630-nm airglow intensity by integrating the volume emission rate along the altitude. So we make two new plots [Fig. (a) and Fig. (b)] to show how the integrated emission rates vary with the increasing neutral temperature and neutral winds, respectively. Fig. (a) shows the result regarding the integrated emission rate as affected by neutral temperature (at -5° geomagnetic latitude on February 1, 2007). The curve in red is fitted as $2^{\text{nd}}$-order polynomial :

$$S = (0.1354 \pm 0.0069)(\Delta T) - (4.6835 \pm 0.2652) \times 10^{-4}(\Delta T)^2 \ ,$$

where S $(\text{km}/(\text{cm}^3 * \text{s}))$ is the change in integrated emission rate and $\Delta T$ (K) is the increase in neutral temperature, compared with the standard conditions of 650 K neutral temperature and zero neutral wind.

[Figure]

$$S = 0.1354 \times \Delta T - 4.6835E(-4) \times (\Delta T)^2$$

Figure (a)

Fig. (b) shows the result regarding the integrated emission rate as affected by neutral wind. The results are obtained based on the same standard conditions as those considered in Fig. (a). The curve in red fits an exponential function :

$$S = (64.8883 \pm 0.7772) \times \{1 - \exp[-(0.0885 \pm 0.0041)(\Delta W)]\} \ ,$$

where $S$ $(km/(cm^3 * s))$ is the change in integrated emission rate and $\Delta W$ (m/s) is the change in neutral wind velocity.

[Figure]

Figure (b)

Therefore, we combine the results of the two fitting functions to approximate the overall change in the integrated emission rate due to the two effects:

$$S = 0.1354(\Delta T) - 4.6835 \times 10^{-4}(\Delta T)^2 + 64.8883[1 - \exp(-0.0885(\Delta W))]$$

Based on the function, we can quantitatively compare the neutral temperature effect with the neutral wind effect. In Fig. (a), the maximum change of the integrated emission rate by increasing the neutral temperature is 9.7859 $(km/(cm^3 * s))$ at 145 K. To get the same changes of the emission rate by varying the neutral wind, it just requires a neutral wind velocity of 1.85 m/s. Above such a velocity, the neutral wind effect would certainly be larger than that of the neutral temperature for this case.

Minor comments:
- Figure 1:Arrows representing wind velocity is not seen clearly.

We have replotted the Fig. 1 as follows, thank you.

[Figure]

- L. 916, Figure 2 –> Figure 3

Our manuscript does not have Line 916. We searched all the "Figure 2" in our article but did not find a similar typo as mentioned. If Referee #1 can still identify the typo, please let us know again. We would revise it. Thank you.

**Anonymous Referee #2**

Chiang et al. demonstrate the influence of meridional wind and neutral temperature to the intensity of 630.0 nm nightglow around the equatorial midnight, altering the SAMI2 model for the resulting plasma density and temperature with the inputs from NRLMSISE-00 for neutral densities and the HWM-93 for neutral wind vectors. The work is potentially interesting and novelty to the community, particularly the finding with respect to the neutral temperature. However, the literature survey by the authors seems to be hasty, the major lacking is that the role of meridional wind to the midnight 630.0 nm airglow enhancement seeing by ISUAL Imager has been studied and published (Rajesh et al.(2014) doi 10.1002/2014JA019927). In addition, the manuscript requires an editing for English before it can be published in the peer-review journal. Given the interesting result and a very valuable dataset, I encourage the authors in extending the content in greater detail that be able to deliver the science finding clearly. Please see further comment below. Summary: Consider for publication after substantial revision Major points:

We thank the Reviewer for reading our article carefully and providing many valuable suggestions for improving the manuscript. We revise the manuscript by taking into account the Reviewer's comments. We also extend the contents and include the observation results in this manuscript in accordance with the Reviewer's suggestions.

(1) Observation data Since the satellite data are used, it would be appropriate to cite Frey et al.(2016) (doi 10.1002/2016JA022616) for the instrument details and the first results of the limb imaging of 630.0 nm airglow using ISUAL by Rajesh et al. (doi 10.1029/2009JA014087 ). The authors put the observation data in the Supplement for some reasons, but it could be nicer if move the section to the main content. The observation data deserve more attention and discussion.

The main purpose of this study is to understand the influence of temperature and meridional neutral wind on the 630.0 nm nightglow by calculating the volume emission rates. The observations by ISUAL can help us realize the tendency in typical solstice condition. In our previous manuscript, we merely wanted to state that our simulations can easily reproduce the selected short-period cases of the brightness patterns observed by ISUAL. But case-study results are not our main points. Considering the observational data that we can access, we suggest that statistical analyses are a more appropriate method to unveil the midnight brightness mechanism.

So in the previous manuscript, we put the observation data in the Supplement. Since Referee #2 thinks that the observation data deserve more attention and discussion, we agree to move them to the main contents in Line 275-286. Moreover, we also add the two references suggested by Referee #2 in Line 53-56 and references section.

(2) The effect of meridional winds to the 630.0 nm midnight brightness By reading this work and Rajesh et al. (2014), I happened to find many similarities in between. Both of the groups modulate the HWM-93 meridional winds on the SAMI2 model and apparently find that the meridional wind utilizes the location and intensity of the airglow brightness. What is the novelty of this work out of Rajesh et al. (2014) in the effect of meridional winds to the midnight brightness? The authors should include the comparison in the content and give the credit to the previous work properly.

We thank the Referee's suggestion. We discuss the differences in detail between the work by Rajesh et al. [2014] and our study. In our manuscript, we include the following discussion in Line 299-315 to compare the two studies.

Rajesh et al. [2014] showed their simulation results and claimed that using merely the background meridional winds could reproduce the observed brightness. They selected a few cases of ISUAL image data and compared those data with the simulation results by the SAMI2 model. Nevertheless, using such a method by Rajesh et al. [2014], one should be very careful about the details when it comes to physical insights or conclusions drawn from the study. This is because ISUAL only provided optical data and there was not any instrument on the satellite to directly observe the relevant conditions (temperature, wind field, etc.) in the environment. Without such observations to provide constraints for modeling, one can easily reproduce similar-looking results of selected short-period data by adjusting modeling parameters in simulations. However, images seemingly similar to that of an ISUAL observation could be produced from simulation results using considerably different parameter values, which may correspond to different dominant mechanisms. Thus, when there are few constraints for the parameter values, roughly comparing a short-period case of ISUAL image data with simulation results without paying attention to details may lead to an interpretation of brightness production mechanisms that is different from the real situation.

The production mechanisms of 630-nm bright spot around midnight from ISUAL observations have been explained by Adachi et al. [2010]. Adachi et al. [2010] suggested the midnight temperature maximum (MTM) effect can well explain the bright spot based on the observation timing and brightness locations. Our previous research (Chiang et al. [2013]) also reached similar conclusions based on statistical studies using two years of ISUAL data. The brightness region tends to appear between the geographic equator and magnetic equator as Fig. 5 in Chiang et al. [2013] indicates (see figure below). This figure shows the sequencing data observed from different longitudinal regions by ISUAL. The dotted red lines indicate the geomagnetic equator; the solid red lines indicate the geographic equator. Rajesh et al. [2014] claimed that the production mechanism of midnight brightness can be explained by meridional winds. The brightness region in their simulation results, however, basically appeared on the winter side of the magnetic equator in the solstices due to the summer-to-winter wind, regardless of where the geographic equator was. Thus, with the consideration of the location of the geographic equator, which is a significant physical factor associated with the MTM effect, the observation results of Fig. 5 in Chiang et al. [2013] indicated that the real situation would actually be different from the case simulated by Rajesh et al. [2014]. Thus the production mechanisms of midnight brightness require different interpretations from those provided by Rajesh et al. [2014].

Thus, we propose that the production of midnight brightness should not be explained by considering merely the effect of meridional neutral wind. Both temperature change and meridional neutral wind can lead to variations of the 630.0 nm nightglow intensity while the latter is more effective. These two effects should be taken into account in the study of midnight brightness.

[Figure]

Note: Fig. 5 in Chiang et al. [2013] shows the observations from three different longitudinal regions [(i), (ii) and (iii)] that correspond to the different declination angles. Orbit (i) was in the longitudinal region (between -15° ~ +150° longitude) where the geomagnetic equator is northward of the geographic equator with the declination angle around 0°. Orbit (iii) was in the region (between -85° ~ -60° longitude) with the geomagnetic equator southward of the geographic equator and the declination angle around 0°. Orbit (ii) was in the geographic region between -60° ~ -15° longitude, with a declination angle around -20° (westward). The solid lines and dashed lines indicate the geographic equator and geomagnetic equator, respectively.

Line 116-117 What is special of O+ density along the magnetic line with apex altitude between 265 and 315 km ? Can you show the model result between altitude 150 to 315 km for all latitude?

Sorry for our typo. We have modified this sentence to "Figure 1 shows the O+ density along the magnetic lines with altitudes between 150 and 315 km in the latitude-altitude plane at the time and longitude described above" in Line 118-120.

Line 214-226 Again, what is the new finding out of fig.3 in Rajesh et al. (2014) ?

Figure 3 in Rajesh et al. [2014] shows statistical results of midnight brightness for different seasons using all the ISUAL images. They collected all the airglow mode data to consider the occurrence of the brightness region but they did not separate the situations for different longitudinal regions.

As we explained in our response to the previous question, Fig. 5 in Chiang et al. [2013] indicates that the latitudinal locations of brightness observed from 3 different orbits (different longitudes) are quite different. We need to consider both temperature change and meridional neutral wind such that the production of midnight brightness in different longitudinal regions can be appropriately addressed. Thus, the statistical results of Fig. 3 in Rajesh et al. [2014] can be considered preliminary work to address the production of midnight brightness, but a broader study to include more relevant physics, such as one also considering the physical factors related to the longitudes, is warranted so as to improve our understanding on the topic. This is also the reason why our study in this manuscript just focuses on the specific longitudinal regions.

Figure 1 has to be modified, what is the reason that the authors didn't convert [O+] density to volume emission rate of 630.0 nm nightglow while the observation images are the airglow intensities?

The effects of neutral wind and temperature on the volume emission rate of the 630.0 nm nightglow are shown in Fig. 2 in our manuscript. The volume emission rate of the 630.0 nm nightglow in the F2 region can be derived as follows:

$$I_{630} = \frac{A_{1D}\mu_D\gamma[O_2][O^+]}{k_1[N_2] + k_2[O_2] + k_3[O] + A_{1D} + A_{2D}}$$

It shows that the volume emission rate is associated with neutral and charged densities. Charged density can be shifted along the field line by neutral wind. On the other hand, most of the items, including charged density, neutral densities and chemical reaction rates, can be affected by temperature variation. Here we would like to explain the thread of thoughts in describing Fig. 1 and Fig. 2. In the context, we first let readers understand the neutral wind effect on charged densities (as shown in Fig. 1), and subsequently we show the effects of neutral wind and temperature on the volume emission rate of the 630.0 nm nightglow (as shown in Fig. 2). Referee #2 suggested that we plot volume emission rate instead of [O+] density in Figure 1. If we plot volume emission rate as suggested, that means both the neutral wind effect and temperature effect need to be considered in Figure 1. Thus it will require lots of figures to show the results because temperature changes need to be considered. We are afraid that readers would be confused by the large number of plots in such an early part of the manuscript, and thus it might not be easy for them to understand our points. Therefore, we tend to keep Fig. 1 as it is shown in the previous manuscript.

Minor points line by line:
Line 43 enhancement > increase

Thank you. We have revised it in Line 43.

Line 45-46 ": : :.first reported the MTM: : : " should be ": : :reported the MTM phenomenon first"

Thank you. We have revised it in Line 45.

Line 61 What are the different mechanisms addressed in Chiang et al. (2013)? The readers would be pleased to learn the relevant work leading by the same author.

Thanks for the Reviewer's comment, we have put the sentence in Line 60-67.

The following figure is Fig. 6 in Chiang et al. [2013]. In the paper, our major goals are to investigate the different patterns of midnight brightness observed by ISUAL and to consider the possible mechanisms for all kinds of cases. Occurrence rates of the four brightness types from all the orbits in each month are shown in the figure: single equatorial brightness (SEB) cases are in green, double equatorial brightness (DEB) in yellow, conjugate brightness (CB) in red, and no brightness (NB) in blue. We found that midnight brightness was controlled by different sources at different locations. First, NB was associated with the ionospheric annual anomaly during May to July. Second, we suppose that SEB and DEB were associated primarily with the MTM effect and the featured temperature variation. Third, the CB case, however, was associated largely with the winter anomaly which the neutral wind plays a role in its formation. It is necessary to take into account the locations and seasons when explaining the mechanisms of midnight brightness occurrence. Overall, the global midnight brightness can be contributed by several effects including the influence of the MTM effect, summer-to-winter neutral wind and ionospheric anomaly.

[Figure]

Line 142-143 Rewrite the sentence please.

Thanks for the Reviewer's comment, we have rewritten the sentence in Line 144-147 as follows:

"In order to explore the effects of temperature change, we modify the codes of SAMI2 by increasing 50 K per run as the inputs, and perform the simulations to calculate the emission intensity values associated with different temperature conditions."

Line 193-195 Rewrite the sentence please.

Thanks for the Reviewer's comment, we have rewritten the sentence in Line 185-187 as follows:

"Therefore, we suggest that the low-latitude emission enhancement in the winter hemisphere be achieved by plasma accumulation brought about by the summer-to-winter neutral wind."

Line 202-203 Rewrite the sentence please.

Thanks for the Reviewer's comment, we have rewritten the sentence in Line 204-207 as follows:

[revised manuscript text omitted]

Figure 2

[Figure]

Figure 3

[Figure]

Figure 4

[Figure]

Figure 5

[Figure]

Figure 6

[Figure]

Figure 7

[Figure]

---

## Referee Report (RR1)

**The 2nd review of "Variation of the 630.0 nm airglow emission with meridional neutral wind and neutral temperature around midnight" by Chiang et al.**

**Summary:** The authors have addressed all my previous concerns thoroughly and the content has been improved distinctively. However, the unit of the integrated emission rate sounds incorrect, and the relevant content is blurry. Given the interesting finding in the turning point of the temperature against the volume emission rate, **this work is worth to consider for publication after the substantial revision.**

According to the explanation in Section 4, I am trying to ,the change ($S_{\Delta T}$ and $S_{\Delta W}$) in the integrated emission rate along the altitude h in the temperature and the neutral wind can be write down as below,

$$S_{\Delta T}(h) = R_2(T_2, h) - R_1(T_1, h) = \int_0^h I(T_2, z)dz - \int_0^h I(T_1, z)dz$$

Where $R_1$ $and$ $R_2$ are the Integrated emission rate with respect to temperature $T_1$ and $T_2$.

$$S_{\Delta W}(h) = R_2(W_2, h) - R_1(W_1, h) = \int_0^h I(W_2, z)dz - \int_0^h I(W_1, z)dz$$

Where $R_1$ $and$ $R_2$ are the Integrated emission rate with respect to neutral wind $W_1$ and $W_2$.

Combine the both temperatures and neutral winds , the change of the integrated emission rate along the altitude h becomes

$$S_{\Delta T, \Delta W}(h) = R_2(T_2, W_2, h) - R_1(T_1, W_1, h) = \int_0^h I(T_2, W_2, z)dz - \int_0^h I(T_1, W_1, z)dz.$$

**Major points:**

1. The unit of the change of the integrated emission rate appears to be incorrect. It should be in the same of the volume emission rate (photons/ $cm^3$/s) multiplied by a length unit, more specifically, km- photons/ $cm^3$/s.
2. Line 264-267:"The maximum change of the integrated emission rate by increasing the neutral temperature is …… at 145 K. " I am confused by the sentence. As my understanding, Figure 4 (a) is the change of the temperature verses the change of the integrated emission rate. However, the sentence is telling me that it is the change of the integrated emission rate in the certain temperature (145 K). Could you elaborate which parameters are actually compared in Figure 4?
3. If my understanding is correct,
   $$S_{\Delta T}(h) = R_2(T_2, h) - R_1(T_1, h) = \int_0^h I(T_2, z)dz - \int_0^h I(T_1, z)dz$$
   We need a fixed h to make $\Delta T$-S plot, but the authors did not mention any altitude dependence with respect to Figure 4,so this is unclear to me what is the physical meaning of Figure 4?

**Minor points:**

The authors used $S$ for all the change of the integrated emission rate despite of it is $\Delta T$ or $\Delta W$ dependent. It is confusing when read it through. I suggest to change the notation in $S_{\Delta T}$, $S_{\Delta W}$ and $S_{\Delta T, \Delta W}$.

End review

July 2018

---

## Referee Report (RR2)

**The 3rd review of "Variation of the 630.0 nm airglow emission with meridional neutral wind and neutral temperature around midnight" by Chiang et al.**

The authors have rearranged the content in a clear way to deliver their finding to the readers. I have no objection to accept the paper as it stands. That's been said, the figure quality can be improved by saving in postscript format.

End review

October 2018

---

## Author Response (AR2)

**Reply to the 1st review report of "Variation of the 630.0 nm airglow emission with meridional neutral wind and neutral temperature around midnight" by Chiang et al.**

Incorporating the previous reviewer's comments, the manuscript has been revised. This paper could contribute to studies of the ionospheric dynamics and disturbances. Consequently, this paper is worth publishing in this journal. However, this reviewer recommends the authors to address the following minor comments.

We would like to thank Referee #1 for recognizing the contribution of our work to studies of the ionospheric dynamics and disturbances. In the revised manuscript we have tried to consider all the suggestions and comments that were raised. Here we reply to the Referee #1's comments accordingly as follows.

**-- Abstract and conclusion:**

It would be better to describe how largely the temperature variation contributes to the airglow intensity variations compared to the effect of the neutral winds.

We thank Referee#1 for providing the suggestions. Based on our estimation, it would require a temperature change of 145 K to produce a change in the integrated emission rate by 9.8 km-photons/cm3/sec, while it only needs the neutral wind velocity to change by 1.85 m/sec to cause the same change in the integrated emission rate. We have added these descriptions in abstract and conclusion session.

**-- LL. 65-67,**

"Cases of global midnight brightness were successfully categorized into four types that were mainly due to the influence of temperature changes, neutral wind and ionospheric anomaly.": The authors mention that there are "four types" at l. 65, but only three types are explained at ll. 65-66. According to Chiang et al. [2013], the remaining one type is "no airglow intensity enhancement". This reviewer recommends the authors to change "four types" to "three types" in this manuscript, and also add a word "enhancement" at the part describing "global midnight brightness" to describe apparently enhancement of the 630-nm airglow intensity.

Thank Referee#1 for providing the suggestion. We have revised it in Line 65. And we also use "enhancement of global midnight brightness" to describe apparently enhancement of the 630-nm airglow intensity in Line 66.

**-- LL. 250 and 257**

Unit of "S" can be described as Rayleigh, defined as a column emission rate of 1010

photons per square meter per column per second.

We thank Referee#1 for providing the suggestion. The Referee#2 raised the unit issue too and suggested that we regard "km-photons/cm3/sec" as the unit of the volume emission rate change ( $\Delta$ S). Because the unit of the intensity in Fig. 2 is "photons/cm3/sec", it is more consistent to consider "km-photons/cm3/sec" the unit of  $\Delta$ S in Fig. 4. We think it is easier for readers to understand them.

-- Figure 3a shows neutral temperature dependence of k\_3[O], k\_1[N2], and k\_2[O2]. It is useful for the reader to describe the temperature dependences of coefficients (k\_3, k\_1, and k\_2) and densities ([O], [N2], and [O2]) separately to show each contribution to the temperature dependence of the volume emission rate. When the neutral temperature increases from 660 K to 900 K, k\_1 and k\_2 decrease by 6% and 4%, respectively, and k\_3 increases by 7%. Therefore, it is found that temperature dependence of the three parameters shown in Figure 3a (k\_3[O], k\_1[N2], and k\_2[O2]) are mainly ascribed to that of the atomic and molecular densities ([O], [N2], and [O2]), and that the coefficients (k\_3, k\_1, and k\_2) does not change significantly.

Thanks for Referee #1's nice suggestions. We have added the new Fig. 3(a) and 3(c) to show the particle densities separately. So in this manuscript the rate coefficients  $(k_1, k_2 \text{ and } k_3)$  and the densities of [O], [N2] and [O2] show each contribution to the temperature dependence of the volume emission rate. We also added new descriptions in the "Results and Analysis" session in Lines 180-186.

-- Last comment of the previous review report
L. 916, Figure 2 --> Figure 3
was a mistake by this reviewer. This reviewer apologize the authors.

We thank Referee #1for reviewing the manuscript.

**Reply to the 2nd review report of "Variation of the 630.0 nm airglow emission with meridional neutral wind and neutral temperature around midnight" by Chiang et al.**

**The 2nd review of "Variation of the 630.0 nm airglow emission with meridional neutral wind and neutral temperature around midnight" by Chiang et al.**

**Summary:** The authors have addressed all my previous concerns thoroughly and the content has been improved distinctively. However, the unit of the integrated emission rate sounds incorrect, and the relevant content is blurry. Given the interesting finding in the turning point of the temperature against the volume emission rate, **this work is worth to consider for publication after the substantial revision.**

According to the explanation in Section 4, I am trying to ,the change  $(S_{\Delta T} \text{ and } S_{\Delta W})$  in the integrated emission rate along the altitude h in the temperature and the neutral wind can be write down as below,

$$S_{\Delta T}(h) = R_2(T_2, h) - R_1(T_1, h) = \int_0^h I(T_2, z) dz - \int_0^h I(T_1, z) dz$$

Where  $R_1$  and  $R_2$  are the Integrated emission rate with respect to temperature  $T_1$  and  $T_2$ .

$$S_{\Delta W}(h) = R_2(W_2, h) - R_1(W_1, h) = \int_0^h I(W_2, z) dz - \int_0^h I(W_1, z) dz$$

Where  $R_1$  and  $R_2$  are the Integrated emission rate with respect to neutral wind  $W_1$  and  $W_2$ . Combine the both temperatures and neutral winds, the change of the integrated emission rate along the altitude h becomes

$$S_{\Delta T,\Delta W}(h) = R_2(T_2, W_2, h) - R_1(T_1, W_1, h) = \int_0^h I(T_2, W_2, z) dz - \int_0^h I(T_1, W_1, z) dz$$

We thank Referee #2 for providing the constructive comments. These comments made by Referee #2 significantly help us improve the explanation of our calculations on the emission rates in different temperature and neutral wind conditions. Therefore, we have incorporated Referee #2's comment in this manuscript (Lines 253-267). We also take into account Referee#2's other comments and revised the manuscript accordingly. Here we reply to the Referee #2's comments accordingly as follows.

**Major points:**

1. The unit of the change of the integrated emission rate appears to be incorrect. It should be in the same of the volume emission rate (photons/  $cm^3/s$ ) multiplied by a length unit, more

specifically, km- photons/  $cm^3/s$ .

We are sorry that we did not write down "photons" in the unit in the previous manuscript. We thank the Referee#2 and we have revised them in this manuscript.

2. Line 264-267: "The maximum change of the integrated emission rate by increasing the neutral temperature is ..... at 145 K." I am confused by the sentence. As my understanding, Figure 4 (a) is the change of the temperature verses the change of the integrated emission rate. However, the sentence is telling me that it is the change of the integrated emission rate in the certain temperature (145 K). Could you elaborate which parameters are actually compared in Figure 4?

We are sorry that our previous sentences about the neutral temperature are not clear enough. The "145 K" in the article is the temperature change ( $\Delta$ T). We have revised the sentences in Lines 289-293.

3. If my understanding is correct,

$$S_{\Delta T}(h) = R_2(T_2, h) - R_1(T_1, h) = \int_0^h I(T_2, z) dz - \int_0^h I(T_1, z) dz$$

We need a fixed h to make  $\Delta T$ -S plot, but the authors did not mention any altitude dependence with respect to Figure 4, so this is unclear to me what is the physical meaning of Figure 4?

As mentioned earlier, we have incorporated Referee #2's comment to improve the explanation of our calculations on the emission rates in different temperature and neutral wind conditions (Lines 253-267). We also provided the altitude information in Lines 268-271. From Fig. 4(a), increasing the neutral temperature by about 145 K leads to the maximum change of the integrated emission rate of 9.7859 km-photons/cm3/sec. In contrast, to get the same changes of the emission rate by varying the neutral wind, it just requires a change of neutral wind velocity by 1.85 m/sec (Fig. 4(b)). With the above estimation, the neutral wind effect would certainly be larger than that of the neutral temperature for this case. These explanations can be found in Lines 289-294.

**Minor points:**

The authors used S for all the change of the integrated emission rate despite of it is  $\Delta T$  or  $\Delta W$  dependent. It is confusing when read it through. I suggest to change the notation in  $S_{\Delta T}$ ,  $S_{\Delta W}$

and  $S_{\Delta T, \Delta W}$ .

Thank Referee #2 for providing the suggestions. They are revised in the manuscript.

| 1 | Variations of the 630.0 nm airglow emission with meridional                                   |
|---|-----------------------------------------------------------------------------------------------|
| 2 | neutral wind and neutral temperature around midnight                                          |
| 3 | Chih-Yu Chiang 1 , Sunny Wing-Yee Tam 1 , Tzu-Fang Chang 1,2 |
| 4 | 1 Institute of Space and Plasma Sciences, National Cheng Kung University, Tainan   |
| 5 | 70101, Taiwan                                                                                 |
| 6 | 2 Institute for Space-Earth Environmental Research, Nagoya University, Nagoya      |
|   |                                                                                               |

7 464-8601, Japan

**8 Abstract**

9 The ISUAL payload onboard the FORMOSAT-2 satellite has often observed 10 airglow bright spots around midnight at equatorial latitudes. Such features had been 11 suggested as the signature of thermospheric midnight temperature maximum (MTM) 12 effect, which was associated with temperature and meridional neutral winds. This 13 study investigates the influence of neutral temperature and meridional neutral wind on 14 the volume emission rates of the 630.0 nm nightglow. We utilize the SAMI2 model to 15 simulate the charged and neutral species at the 630.0 nm nightglow emission layer 16 under different temperatures with and without the effect of neutral wind. The results 17 show that the neutral wind is more efficient than temperature variation in affecting the 18 nightglow emission rates. For example, based on our estimation, it would require 19 a-temperature change of 145 K to produce a change in the integrated emission rate by 20 9.8 km-photons/cm3/sec, while it only needs the neutral wind velocity to change by 21 1.85 m/sec to cause the same change in the integrated emission rate. 
[revised manuscript text omitted]
  $-5^{\circ}$ 173 geomagnetic latitude on February 1, 2007. In Fig. 3(a), we plot [O], [N2] and [O2] in dotted, dashed and solid lines respectively. Then the corresponding loss rates of these 174 neutral species are shown in Fig. 3(b). In Fig. 3(c),  $[O^+]$  with and without neutral wind 175 176 effect are plotted with dotted line and solid line respectively. The values of  $\gamma$ 177  $[O^+][O_2]$ , which are related to the production rate and in the numerator of Eq. (4), are

plotted in Fig. 3(d). The dotted line represents the normal neutral wind condition, andthe solid line for the windless condition.

| 180 | When the neutral temperature increases from 600 K to 900 K, the rate                         |
|-----|----------------------------------------------------------------------------------------------|
| 181 | coefficients $k_1$ and $k_2$ decrease by 5.8% and 3.7%, respectively, and $k_3$ increases by |
| 182 | 7.4% as calculated from Table 1. The rate coefficients $k_1$ , $k_2$ and $k_3$ do not change |
| 183 | significantly. However, in the same temperature range, $[O]$ , $[N_2]$ and $[O_2]$ show      |
| 184 | prominent increases of 253%, 363% and 171%, respectively, as shown in Fig. 3(a).             |
| 185 | Therefore, the atomic and molecular densities dominate the changes of the loss rates         |
| 186 | (Fig. 3(b)).                                                                                 |

187

**188 **4. Discussion**

[revised manuscript text omitted]

In order to quantitatively describe the effects of neutral temperature and meridional neutral winds, we calculate the 630-nm airglow intensity by integrating the volume emission rate along the altitude. Thus, the change in the integrated emission 254 rate  $(\Delta S_T)$  over a fixed altitude range h1 to h2 due to a change in temperature from

 $T_1$  to  $T_2$  can be written as: 255

256
$$\Delta S_T = S(T_2, W) - S(T_1, W) = \int_{h_1}^{h_2} I_{630}(T_2, W, z) dz - \int_{h_1}^{h_2} I_{630}(T_1, W, z) dz , \qquad (5)$$

257 where S is the integrated emission rate from height h1 to h2 as a function of temperature and neutral wind speed W. Similarly, the change in the integrated 258 emission rate ( $\Delta S_w$ ) over a fixed altitude range h1 to h2 due to a change in the 259 260

neutral wind speed from  $W_1$  to  $W_2$  can be obtained as:

261
$$\Delta S_W = S(T, W_2) - S(T, W_1) = \int_{h_1}^{h_2} I_{630}(T, W_2, z) dz - \int_{h_1}^{h_2} I_{630}(T, W_1, z) dz , \qquad (6)$$

262 Combining the changes in both temperature and neutral wind, one may express the

263 change of the integrated emission rate over the altitude range as:

264
$$\Delta S_{T,W} = S(T_2, W_2) - S(T_1, W_1) = \int_{h_1}^{h_2} I_{630}(T_2, W_2, z) dz - \int_{h_1}^{h_2} I_{630}(T_1, W_1, z) dz$$

265 One can show that to the leading order, the above equation reduces to

$$\Delta S_{T,W} = \Delta S_T + \Delta S_W , \qquad (7)$$

with  $\Delta S_T$  in Eq. (5) evaluated at  $W = W_1$  and  $\Delta S_W$  in Eq. (6) evaluated at  $T = T_1$ . 267 Based on Eq. (4), we calculated  $I_{630}$  for different temperatures and neutral wind 268 conditions, and then according to the integrals in Eq. (5) and (6), integrated the 269 270 emission rates over the major altitudes of the 630.0 nm nightglow emission layer, 271 ranging from 150 to 315 km altitude. Figure 4(a) and 4(b) show how the integrated 272 emission rates vary with the increases in the neutral temperature and neutral wind speed, respectively. Fig. 4(a) shows the result regarding the integrated emission rate as
affected by neutral temperature (at -5° geomagnetic latitude on February 1, 2007). The
curve in red is fitted as 2nd-order polynomial :

276
$$\Delta S_T = (0.1354 \pm 0.0069)(\Delta T) - (4.6835 \pm 0.2652) \times 10^{-4} (\Delta T)^2 ,$$

where  $\Delta S_T$  (km-photons/cm3/sec) is the change in integrated emission rate and  $\Delta T$  (K) is the increase in neutral temperature, compared with the standard conditions of 650 K neutral temperature and zero neutral wind. Fig. 4(b) shows the result regarding the integrated emission rate as affected by neutral wind. The results are obtained based on the same standard conditions as those considered in Fig. 4(a). The curve in red fits an exponential function :

283
$$\Delta S_W = (64.8883 \pm 0.7772) \times \{1 - \exp[-(0.0885 \pm 0.0041)(\Delta W)]\}$$

[revised manuscript text omitted]
(^{1}D) + O \rightarrow O + O$         | $k_3 = (3.73 + 1.1965 \times 10^{-1} \text{ T}_n^{0.5} - 6.5898 \times 10^{-4} \text{ T}_n) \times 10^{-12}$                                                                                                                                                                                                                                                                                                                                                                                                                                                                                                                                                                                                                                                                                                                                                                                                                                                                                                                                                                                                                                                                                                                                                                                                                                                                                                                                                                                                                                                                                                                                                                                                                                                                                                                                                                                                                                                                                                                                                                                                                                       |
|      | $O(^{1}D) \rightarrow O + hv(630.0nm)$   | $A_{1D} = 7.1 \times 10^{-3}$                                                                                                                                                                                                                                                                                                                                                                                                                                                                                                                                                                                                                                                                                                                                                                                                                                                                                                                                                                                                                                                                                                                                                                                                                                                                                                                                                                                                                                                                                                                                                                                                                                                                                                                                                                                                                                                                                                                                                                                                                                                                                                                      |
|      | $O(^{1}D) \rightarrow O + hv(634.4nm)$   | $A_{2D} = 2.2 \times 10^{-3}$                                                                                                                                                                                                                                                                                                                                                                                                                                                                                                                                                                                                                                                                                                                                                                                                                                                                                                                                                                                                                                                                                                                                                                                                                                                                                                                                                                                                                                                                                                                                                                                                                                                                                                                                                                                                                                                                                                                                                                                                                                                                                                                      |
| 559  | Note: $T_{eff} = 0.67T_i + 0.33T_n$ (T   | $_{eff}$ : effective temperature, $T_i$ : ion temperature, $T_n$ : neutral                                                                                                                                                                                                                                                                                                                                                                                                                                                                                                                                                                                                                                                                                                                                                                                                                                                                                                                                                                                                                                                                                                                                                                                                                                                                                                                                                                                                                                                                                                                                                                                                                                                                                                                                                                                                                                                                                                                                                                                                                                                                         |
| 560  | temperature) [StMaurice and              | Torr, 1978]                                                                                                                                                                                                                                                                                                                                                                                                                                                                                                                                                                                                                                                                                                                                                                                                                                                                                                                                                                                                                                                                                                                                                                                                                                                                                                                                                                                                                                                                                                                                                                                                                                                                                                                                                                                                                                                                                                                                                                                                                                                                                                                                        |
| 561  |                                          |                                                                                                                                                                                                                                                                                                                                                                                                                                                                                                                                                                                                                                                                                                                                                                                                                                                                                                                                                                                                                                                                                                                                                                                                                                                                                                                                                                                                                                                                                                                                                                                                                                                                                                                                                                                                                                                                                                                                                                                                                                                                                                                                                    |
| 501  |                                          |                                                                                                                                                                                                                                                                                                                                                                                                                                                                                                                                                                                                                                                                                                                                                                                                                                                                                                                                                                                                                                                                                                                                                                                                                                                                                                                                                                                                                                                                                                                                                                                                                                                                                                                                                                                                                                                                                                                                                                                                                                                                                                                                                    |
| 562  |                                          |                                                                                                                                                                                                                                                                                                                                                                                                                                                                                                                                                                                                                                                                                                                                                                                                                                                                                                                                                                                                                                                                                                                                                                                                                                                                                                                                                                                                                                                                                                                                                                                                                                                                                                                                                                                                                                                                                                                                                                                                                                                                                                                                                    |
| 002  |                                          |                                                                                                                                                                                                                                                                                                                                                                                                                                                                                                                                                                                                                                                                                                                                                                                                                                                                                                                                                                                                                                                                                                                                                                                                                                                                                                                                                                                                                                                                                                                                                                                                                                                                                                                                                                                                                                                                                                                                                                                                                                                                                                                                                    |
| 563  |                                          |                                                                                                                                                                                                                                                                                                                                                                                                                                                                                                                                                                                                                                                                                                                                                                                                                                                                                                                                                                                                                                                                                                                                                                                                                                                                                                                                                                                                                                                                                                                                                                                                                                                                                                                                                                                                                                                                                                                                                                                                                                                                                                                                                    |
|      |                                          |                                                                                                                                                                                                                                                                                                                                                                                                                                                                                                                                                                                                                                                                                                                                                                                                                                                                                                                                                                                                                                                                                                                                                                                                                                                                                                                                                                                                                                                                                                                                                                                                                                                                                                                                                                                                                                                                                                                                                                                                                                                                                                                                                    |
| 564  |                                          |                                                                                                                                                                                                                                                                                                                                                                                                                                                                                                                                                                                                                                                                                                                                                                                                                                                                                                                                                                                                                                                                                                                                                                                                                                                                                                                                                                                                                                                                                                                                                                                                                                                                                                                                                                                                                                                                                                                                                                                                                                                                                                                                                    |
|      |                                          |                                                                                                                                                                                                                                                                                                                                                                                                                                                                                                                                                                                                                                                                                                                                                                                                                                                                                                                                                                                                                                                                                                                                                                                                                                                                                                                                                                                                                                                                                                                                                                                                                                                                                                                                                                                                                                                                                                                                                                                                                                                                                                                                                    |
| 565  |                                          |                                                                                                                                                                                                                                                                                                                                                                                                                                                                                                                                                                                                                                                                                                                                                                                                                                                                                                                                                                                                                                                                                                                                                                                                                                                                                                                                                                                                                                                                                                                                                                                                                                                                                                                                                                                                                                                                                                                                                                                                                                                                                                                                                    |
|      |                                          |                                                                                                                                                                                                                                                                                                                                                                                                                                                                                                                                                                                                                                                                                                                                                                                                                                                                                                                                                                                                                                                                                                                                                                                                                                                                                                                                                                                                                                                                                                                                                                                                                                                                                                                                                                                                                                                                                                                                                                                                                                                                                                                                                    |
| 566  |                                          |                                                                                                                                                                                                                                                                                                                                                                                                                                                                                                                                                                                                                                                                                                                                                                                                                                                                                                                                                                                                                                                                                                                                                                                                                                                                                                                                                                                                                                                                                                                                                                                                                                                                                                                                                                                                                                                                                                                                                                                                                                                                                                                                                    |
|      |                                          |                                                                                                                                                                                                                                                                                                                                                                                                                                                                                                                                                                                                                                                                                                                                                                                                                                                                                                                                                                                                                                                                                                                                                                                                                                                                                                                                                                                                                                                                                                                                                                                                                                                                                                                                                                                                                                                                                                                                                                                                                                                                                                                                                    |
| 567  |                                          |                                                                                                                                                                                                                                                                                                                                                                                                                                                                                                                                                                                                                                                                                                                                                                                                                                                                                                                                                                                                                                                                                                                                                                                                                                                                                                                                                                                                                                                                                                                                                                                                                                                                                                                                                                                                                                                                                                                                                                                                                                                                                                                                                    |
|      |                                          |                                                                                                                                                                                                                                                                                                                                                                                                                                                                                                                                                                                                                                                                                                                                                                                                                                                                                                                                                                                                                                                                                                                                                                                                                                                                                                                                                                                                                                                                                                                                                                                                                                                                                                                                                                                                                                                                                                                                                                                                                                                                                                                                                    |
| 568  |                                          |                                                                                                                                                                                                                                                                                                                                                                                                                                                                                                                                                                                                                                                                                                                                                                                                                                                                                                                                                                                                                                                                                                                                                                                                                                                                                                                                                                                                                                                                                                                                                                                                                                                                                                                                                                                                                                                                                                                                                                                                                                                                                                                                                    |
| 5(0) |                                          |                                                                                                                                                                                                                                                                                                                                                                                                                                                                                                                                                                                                                                                                                                                                                                                                                                                                                                                                                                                                                                                                                                                                                                                                                                                                                                                                                                                                                                                                                                                                                                                                                                                                                                                                                                                                                                                                                                                                                                                                                                                                                                                                                    |
| 209  |                                          |                                                                                                                                                                                                                                                                                                                                                                                                                                                                                                                                                                                                                                                                                                                                                                                                                                                                                                                                                                                                                                                                                                                                                                                                                                                                                                                                                                                                                                                                                                                                                                                                                                                                                                                                                                                                                                                                                                                                                                                                                                                                                                                                                    |
| 570  |                                          |                                                                                                                                                                                                                                                                                                                                                                                                                                                                                                                                                                                                                                                                                                                                                                                                                                                                                                                                                                                                                                                                                                                                                                                                                                                                                                                                                                                                                                                                                                                                                                                                                                                                                                                                                                                                                                                                                                                                                                                                                                                                                                                                                    |
| 570  |                                          |                                                                                                                                                                                                                                                                                                                                                                                                                                                                                                                                                                                                                                                                                                                                                                                                                                                                                                                                                                                                                                                                                                                                                                                                                                                                                                                                                                                                                                                                                                                                                                                                                                                                                                                                                                                                                                                                                                                                                                                                                                                                                                                                                    |
| 571  |                                          |                                                                                                                                                                                                                                                                                                                                                                                                                                                                                                                                                                                                                                                                                                                                                                                                                                                                                                                                                                                                                                                                                                                                                                                                                                                                                                                                                                                                                                                                                                                                                                                                                                                                                                                                                                                                                                                                                                                                                                                                                                                                                                                                                    |
| 011  |                                          |                                                                                                                                                                                                                                                                                                                                                                                                                                                                                                                                                                                                                                                                                                                                                                                                                                                                                                                                                                                                                                                                                                                                                                                                                                                                                                                                                                                                                                                                                                                                                                                                                                                                                                                                                                                                                                                                                                                                                                                                                                                                                                                                                    |
| 572  |                                          |                                                                                                                                                                                                                                                                                                                                                                                                                                                                                                                                                                                                                                                                                                                                                                                                                                                                                                                                                                                                                                                                                                                                                                                                                                                                                                                                                                                                                                                                                                                                                                                                                                                                                                                                                                                                                                                                                                                                                                                                                                                                                                                                                    |
|      |                                          |                                                                                                                                                                                                                                                                                                                                                                                                                                                                                                                                                                                                                                                                                                                                                                                                                                                                                                                                                                                                                                                                                                                                                                                                                                                                                                                                                                                                                                                                                                                                                                                                                                                                                                                                                                                                                                                                                                                                                                                                                                                                                                                                                    |
| 573  |                                          |                                                                                                                                                                                                                                                                                                                                                                                                                                                                                                                                                                                                                                                                                                                                                                                                                                                                                                                                                                                                                                                                                                                                                                                                                                                                                                                                                                                                                                                                                                                                                                                                                                                                                                                                                                                                                                                                                                                                                                                                                                                                                                                                                    |
|      |                                          |                                                                                                                                                                                                                                                                                                                                                                                                                                                                                                                                                                                                                                                                                                                                                                                                                                                                                                                                                                                                                                                                                                                                                                                                                                                                                                                                                                                                                                                                                                                                                                                                                                                                                                                                                                                                                                                                                                                                                                                                                                                                                                                                                    |

**574 Figure Captions**

| 575 | Figure 1. Oxygen ion density plotted in the latitude-altitude plane at 23:00 LT on                                       |
|-----|--------------------------------------------------------------------------------------------------------------------------|
| 576 | February 1, 2007 (left panels) and August 1, 2007 (right panels) in the Asian                                            |
| 577 | region (100°E longitude) from the SAMI-2 model: (a) without neutral wind; (b)                                            |
| 578 | with the effect of normal neutral wind, whose strength and directions are                                                |
| 579 | indicated by the arrows.                                                                                                 |
| 580 | Figure 2. The results of 630.0 nm emission rate at 23 LT at different temperatures and                                   |
| 581 | under different neutral wind conditions for (a) February 1, 2007 and (b) August                                          |
| 582 | 1, 2007: left and right panels respectively for $-5^{\circ}$ and $+5^{\circ}$ geomagnetic latitude;                      |
| 583 | the letters, A, B, C, D and E, for the altitudes of 220 km, 230 km, 240 km, 250                                          |
| 584 | km and 260 km, respectively; for normal neutral wind effect (black dotted lines)                                         |
| 585 | and windless conditions (red solid lines). The neutral wind conditions of Fig. 2                                         |
| 586 | are the same as those shown in Fig. 1.                                                                                   |
| 587 | Figure 3. The profiles of neutral and charged species versus temperature which are                                       |
| 588 | involved in Eq. (4) at 230 km altitudes and $-5^{\circ}$ geomagnetic latitudes on                                        |
| 589 | February 1, 2007. (a) [O], $[N_2]$ and $[O_2]$ versus temperature. (b) The loss rate                                     |
| 590 | terms of $k_1[O]$ , $k_2$ [N 2 ] and $k_3$ [O 2 ] versus temperature. (c) [O + ] versus |
| 591 | temperature with/without the neutral wind effect. (d) The production                                                     |
| 592 | rate-associated term of $\gamma$ [O + ][O 2 ] versus temperature with/without the neutral          |

**593 wind effect.**

| 594 | Figure 4. Quantitative results for how (a) the neutral temperature and (b) the neutral |
|-----|----------------------------------------------------------------------------------------|
| 595 | wind affect the 630-nm airglow intensity.                                              |
| 596 | Figure 5. Plots of the emission rates against the turning temperature between 220-260  |
| 597 | km altitudes.                                                                          |
| 598 | Figure 6. Four observation cases by ISUAL in February 2007 and August 2007 (the        |
| 599 | same periods as shown in Fig. 1).                                                      |
| 600 | Figure 7. ISUAL data in the specific regions and seasons considered in the             |
| 601 | simulations: the nightglow bright spots in valid observation days during (a)           |
| 602 | January-February and (b) July-August.                                                  |
| 603 |                                                                                        |
| 604 |                                                                                        |
| 605 |                                                                                        |
| 606 |                                                                                        |
| 607 |                                                                                        |
| 608 |                                                                                        |
| 609 |                                                                                        |
| 610 |                                                                                        |
| 611 |                                                                                        |